# E7050 Suppresses the Growth of Multidrug-Resistant Human Uterine Sarcoma by Inhibiting Angiogenesis via Targeting of VEGFR2-Mediated Signaling Pathways

**DOI:** 10.3390/ijms24119606

**Published:** 2023-05-31

**Authors:** Tsung-Teng Huang, Chuan-Mu Chen, Song-Shu Lin, Ying-Wei Lan, Hsu-Chen Cheng, Kong-Bung Choo, Ching-Chiung Wang, Tse-Hung Huang, Kowit-Yu Chong

**Affiliations:** 1School of Pharmacy, College of Pharmacy, Taipei Medical University, Taipei 11031, Taiwan; huanghongde@gmail.com (T.-T.H.); crystal@tmu.edu.tw (C.-C.W.); 2Traditional Herbal Medicine Research Center, Taipei Medical University Hospital, Taipei 11031, Taiwan; 3Department of Medical Biotechnology and Laboratory Science, College of Medicine, Chang Gung University, Taoyuan 33302, Taiwan; 4Graduate Institute of Biomedical Sciences, Division of Biotechnology, College of Medicine, Chang Gung University, Taoyuan 33302, Taiwan; 5Department of Life Sciences, Agricultural Biotechnology Center, National Chung Hsing University, Taichung 40227, Taiwan; chchen1@dragon.nchu.edu.tw (C.-M.C.); hcheng@dragon.nchu.edu.tw (H.-C.C.); 6The iEGG and Animal Biotechnology Center and the Rong Hsing Research Center for Translational Medicine, National Chung Hsing University, Taichung 40227, Taiwan; 7Department of Nursing, Chang Gung University of Science and Technology, Taoyuan 33302, Taiwan; lss1192001@yahoo.com.tw; 8Hyperbaric Oxygen Medical Research Laboratory, Bone and Joint Research Center, Linkou Chang Gung Memorial Hospital, Taoyuan 33305, Taiwan; 9Division of Pulmonary Biology, The Perinatal Institute of Cincinnati Children’s Research Foundation, Cincinnati, OH 45229, USA; ying-wei.lan@cchmc.org; 10Centre for Stem Cell Research, Faculty of Medicine and Health Sciences, Universiti Tunku Abdul Rahman, Kajang 43000, Selangor, Malaysia; chookb@utar.edu.my; 11Department of Traditional Chinese Medicine, Linkou Chang Gung Memorial Hospital, Taoyuan 33305, Taiwan; tsehunghuang_1089@yahoo.com.tw

**Keywords:** E7050, VEGFR2, VEGF, multidrug-resistant human uterine sarcoma, angiogenesis

## Abstract

E7050 is an inhibitor of VEGFR2 with anti-tumor activity; however, its therapeutic mechanism remains incompletely understood. In the present study, we aim to evaluate the anti-angiogenic activity of E7050 in vitro and in vivo and define the underlying molecular mechanism. It was observed that treatment with E7050 markedly inhibited proliferation, migration, and capillary-like tube formation in cultured human umbilical vein endothelial cells (HUVECs). E7050 exposure in the chick embryo chorioallantoic membrane (CAM) also reduced the amount of neovessel formation in chick embryos. To understand the molecular basis, E7050 was found to suppress the phosphorylation of VEGFR2 and its downstream signaling pathway components, including PLCγ1, FAK, Src, Akt, JNK, and p38 MAPK in VEGF-stimulated HUVECs. Moreover, E7050 suppressed the phosphorylation of VEGFR2, FAK, Src, Akt, JNK, and p38 MAPK in HUVECs exposed to MES-SA/Dx5 cells-derived conditioned medium (CM). The multidrug-resistant human uterine sarcoma xenograft study revealed that E7050 significantly attenuated the growth of MES-SA/Dx5 tumor xenografts, which was associated with inhibition of tumor angiogenesis. E7050 treatment also decreased the expression of CD31 and p-VEGFR2 in MES-SA/Dx5 tumor tissue sections in comparison with the vehicle control. Collectively, E7050 may serve as a potential agent for the treatment of cancer and angiogenesis-related disorders.

## 1. Introduction

Angiogenesis is characterized by the formation of new capillary blood vessels from the preexisting vasculature. It contributes to various physiological processes, including embryonic growth, fetal development, reproduction, postnatal development, and wound healing [1]. On the other hand, it also plays an important role in pathological conditions, such as tumor growth, invasion, and metastasis [2,3]. Tumor cells can secrete several proangiogenic factors, stimulating endothelial cell migration, proliferation, and capillary-like tube formation; the resulting neovasculature supports tumor growth and metastasis by supplying oxygen and nutrients and excreting carbon dioxide and metabolic wastes [2,4]. Therefore, blocking tumor angiogenesis is an attractive strategy of oncotherapy, which may help inhibit the growth and spread of tumors. To date, numerous growth factors and cytokines are involved in tumor-associated angiogenesis, among which vascular endothelial growth factor (VEGF) is the most essential and specific mediator [5].

VEGF augments most steps of angiogenesis, mainly through binding to receptor tyrosine kinase VEGF receptor 2 (VEGFR2, also known as KDR in humans and Flk-1 in mice) on the surface of endothelial cells [6]. The binding of VEGF to VEGFR2 results in dimerization and autophosphorylation of the receptor and the activation of multiple downstream intracellular protein kinases, such as phosphoinositide 3-kinase/protein kinase B (PI3K/Akt), Src family kinases, phospholipase C gamma (PLCγ), focal adhesion kinase (FAK), extracellular signal-regulated kinases 1 and 2 (ERK1/2), c-Jun N-terminal kinase/stress-activated protein kinase (JNK/SAPK), p38 mitogen-activated protein kinase (MAPK), and heat shock protein (HSP) 27, which have been implicated in cell proliferation, permeability, migration, survival, and microtubule formation [7,8,9]. Accordingly, inhibition of VEGF production by tumor cells and the targeting of VEGFR2 and VEGFR2-mediated signaling pathways to prevent tumor-induced angiogenesis through the use of chemotherapeutic drugs have been considered effective approaches for cancer therapy [10,11,12].

Uterine sarcoma is a highly aggressive and lethal gynecologic malignancy. Treatment of uterine sarcoma remains a major challenge due to its rarity, unknown etiology, and highly divergent genetic aberration [13]. Previous studies have indicated that VEGF is strongly expressed in uterine carcinosarcoma, and tumor angiogenesis in uterine carcinosarcoma may be influenced by VEGF [14]. Based on our previous research, high expression levels of cellular and secreted VEGF are observed in multidrug-resistant human uterine sarcoma MES-SA/Dx5 cells [15]. Anti-angiogenic agents such as bevacizumab and tyrosine kinase inhibitors such as sorafenib and pazopanib have been applied clinically to treat uterine sarcoma [16]. Nevertheless, these treatments showed low efficacy and unexpected toxicity, suggesting that discovery and development of other effective tumor angiogenesis inhibitors with low toxicity are still required.

E7050 has attracted considerable attention recently because of its potent anti-tumor activity. Previous studies have demonstrated that E7050 can significantly inhibit the xenograft growth of various human tumor cell lines, including lung and gastric tumors [17,18]. In addition, E7050 was reported to inhibit phosphorylation of c-Met and VEGFR2 and repress tumor growth and angiogenesis [17]. E7050 treatment also has the potency to prolong the lifespan of tumor-bearing mice without any adverse effects [17]. However, the molecular mechanism underlying the anti-angiogenic effect of E7050 has not been illustrated well.

In this study, we systematically explored the anti-angiogenic effects of E7050 in vitro and in vivo. Treatment with E7050 inhibited angiogenesis in cultured human endothelial cells, as indicated by our cell proliferation, wound healing migration, and tube formation assay results in vitro and our chick embryo CAM assay results in vivo. In addition, through the use of a multidrug-resistant human uterine sarcoma MES-SA/Dx5 cell line-derived xenograft model, E7050 was further shown to suppress tumor growth and tumor-associated angiogenesis. In terms of the underlying molecular mechanisms, the inhibitory effect of E7050 on angiogenesis was associated with the blockade of VEGFR2 activation and its downstream FAK, Src, Akt, JNK, and p38 MAPK signal transduction. Converging these findings, we suggest that E7050 may be further developed as a therapeutic agent for angiogenesis-related diseases.

## 2. Results

### 2.1. E7050 Inhibits the Proliferation, Migration, and Tube Formation of HUVECs

To assess the anti-angiogenic property of E7050 in vitro, we first examined the effects of E7050 on the proliferation of HUVECs using the MTT assay. Results showed that the treatment of cells with E7050 at 5 μM for 6 h had no evident effect on the proliferation of HUVECs when compared to the vehicle control; however, a substantial reduction in the proliferation of HUVECs was observed after treatment with E7050 at concentrations ≥10 μM (Figure 1A). Moreover, the proliferation of HUVECs was remarkably inhibited following 24 h of treatment with E7050 at concentrations ranging from 5 to 25 μM (Figure 1B), indicating that E7050 could decrease the proliferation of HUVECs in vitro. Given the fact that cell migration is one of the key steps in the initiation of angiogenesis [19], a wound healing assay was performed to examine the effect of E7050 on the migration of endothelial cells. As shown in Figure 1C,E, E7050 treatment for 16 h markedly suppressed the migration of HUVECs in a concentration-dependent manner according to the gap distance of scratches; in particular, the migration ability of HUVECs was almost inhibited at the E7050 concentration of 10 μM. Next, we explored the effect of E7050 on the formation of capillary-like tubular structures by endothelial cells, as this is a critical process during angiogenesis [20]. HUVECs were cultivated onto Matrigel in the presence of E7050 at various concentrations (5, 10, and 25 μM), and capillary-like tube formation was examined. HUVECs showed prominent and well-formed tubular structures in the vehicle control group, but exposure to E7050 significantly inhibited the formation of capillary-like tubular structures in a concentration-dependent manner (Figure 1D); almost 60% of capillary-like tubular structures were abolished when HUVECs were incubated with E7050 at the concentration of 25 μM for 6 h (Figure 1F). Taken together, these findings suggest that E7050 displays the anti-angiogenic effect through inhibition of proliferation, migration, and microtubule development in endothelial cells.

### 2.2. E7050 Suppresses Angiogenesis In Vivo

To determine the potential effect of E7050 on angiogenesis in vivo, a well-established angiogenesis model, the chick embryo CAM assay, was also conducted. As illustrated in Figure 2A, E7050 treatment significantly reduced chicken embryonic blood vessel formation in a concentration-dependent manner, whereas the vehicle control group had normal vascularization. Quantification of angiogenesis level based on vessel network lengths showed that E7050 profoundly inhibited neovascularization compared with the vehicle control group (Figure 2B). The above results confirm the noticeable inhibitory function of E7050 on angiogenesis in vivo.

### 2.3. E7050 Suppresses the Phosphorylation of VEGFR2 and the Activation of VEGFR2-Mediated Signaling Pathways in VEGF-Stimulated HUVECs

VEGFR2 is thought to be the primary receptor in VEGF-induced signaling pathways that are responsible for endothelial cell survival, proliferation, migration, tube formation, and angiogenesis [21]. To investigate the potential mechanisms involved in the anti-angiogenic function of E7050, we thus examined the impact of E7050 on the expression level of tyrosine phosphorylation of VEGFR2 in HUVECs stimulated by VEGF. Expression of p-VEGFR2 at Tyr1175 and Tyr996 and total VEGFR2 was detected by Western blot analysis. As shown in Figure 3A,B and Appendix A, the expression levels of VEGFR2 phosphorylation at Tyr1175 and Tyr996 were apparently increased after VEGF stimulation. However, E7050 pretreatment almost completely inhibited the phosphorylation of VEGFR2 at Tyr1175. E7050 also markedly suppressed the phosphorylation of VEGFR2 at Tyr996 in a concentration-dependent manner, suggesting that the anti-angiogenic effect of E7050 may be partially dependent on the inhibition of VEGFR2 activation. Additionally, to confirm the molecular mechanisms by which E7050 inhibits VEGF-induced angiogenesis, we further examined several pivotal components that act downstream of VEGFR2 activation. Notably, E7050 treatment dramatically inhibited the expression levels of p-PLCγ1, p-FAK, p-Src, p-Akt, p-JNK, and p-p38 MAPK in a concentration-dependent manner, while there were no obvious changes in the expression levels of total PLCγ1, FAK, Src, Akt, JNK, and p38 MAPK after the stimulation of VEGF in the presence of 5 and 10 μM E7050 (Figure 3C–J). In contrast, an E7050 concentration of up to 10 μM exhibited no inhibitory effect on VEGF-induced phosphorylation of ERK1/2 and HSP27 (Appendix A). These results indicate that E7050 is able to inhibit angiogenesis through regulating the VEGFR2-mediated signaling pathways in endothelial cells.

### 2.4. E7050 Has No Inhibitory Effect on VEGF Production in MES-SA/Dx5 Cells

To explore whether the anti-angiogenic effect of E7050 involves the modulation of VEGF expression in multidrug-resistant human uterine sarcoma MES-SA/Dx5 cells, VEGF expression in MES-SA/Dx5 cells in response to E7050 treatment was examined. MES-SA/Dx5 cells were treated with E7050 for 24 h, and VEGF protein levels were determined using Western blotting analysis. As shown in Figure 4A,B, VEGF expression levels in cells treated with E7050 at 5 and 10 μM were similar to those of control cells treated with the DMSO vehicle, whereas E7050 at 25 μM significantly upregulated VEGF protein expression. In contrast, the ELISA data showed that E7050 treatment did not affect the secretion of VEGF in cell culture media (Figure 4C). These results suggest that the anti-angiogenic action of E7050 is not mediated by the inhibition of VEGF production.

### 2.5. E7050 Attenuates MES-SA/Dx5 Cells-Derived Conditioned Medium (CM)-Induced Phosphorylation of VEGFR2 and Its Downstream Signaling Mediators in HUVECs

The angiogenic growth factor VEGF released from cancer cells reacts with its specific receptor, VEGFR2, on vascular endothelial cells, which is required for tumor-induced angiogenesis [7]. Therefore, in order to elucidate the mechanism underlying the anti-angiogenic effect of E7050 in MES-SA/Dx5 cells-derived CM-treated HUVECs, we performed Western blot assays to examine whether E7050 could inhibit the expression and phosphorylation of VEGFR2, as well as its downstream signaling pathway components. It was shown that E7050 greatly attenuated MES-SA/Dx5 CM-induced phosphorylation of VEGFR2 in a concentration-dependent manner in HUVECs (Figure 5A,B). We also examined the phosphorylation status of FAK, Src, Akt, JNK, and p38, which are essential protein kinases involved in VEGFR2-mediated signaling. As shown in Figure 5A,C–H, the expression levels of p-FAK, p-Akt, p-JNK, and p-p38 MAPK were markedly increased by MES-SA/Dx5-CM treatment, while the expression level of p-Src was not clearly altered by MES-SA/Dx5-CM treatment. Moreover, E7050 strongly suppressed the expression levels of p-Src, p-Akt, p-JNK, and p-p38 MAPK in MES-SA/Dx5 CM-stimulated HUVECs, but the expression level of p-FAK was slightly reduced by E7050. Overall, these findings imply that E7050 inhibits tumor angiogenesis through suppression of VEGFR2 activation and the blocking of VEGFR2-mediated signaling pathways, which effectively contributes to its anti-cancer activity.

### 2.6. E7050 Inhibits Tumor Growth and Angiogenesis in the MES-SA/Dx5 Cell Line-Derived Xenograft Mouse Model

In order to evaluate whether E7050 inhibits tumor growth by suppressing tumor angiogenesis, an MES-SA/Dx5 tumor xenograft model was set up. As shown in Figure 6, H&E staining was first performed to examine histopathological changes in xenograft tumors from each group. The results showed that E7050 administration effectively inhibited tumor growth, and tumor size was obviously smaller than it was in the vehicle control group. In addition, tumor tissue sections from the xenograft mouse model were stained with specific antibodies against CD31, VEGF, and p-VEGFR2 (Tyr1175). The results of immunohistochemical staining revealed that E7050 treatment evidently decreased the expression of CD31, a marker of microvessel density, in MES-SA/Dx5 xenograft tumors when compared to the vehicle-treated group. Meanwhile, there was no significant difference in VEGF expression in the vehicle-treated group compared with E7050-treated groups. However, E7050 treatment pronouncedly reduced the expression of p-VEGFR2 in tumor tissue sections, further demonstrating that E7050 played an important role in suppressing angiogenesis at least partly through blocking of the activation of VEGFR2. These results indicate that the anti-tumor effect of E7050 is closely related to its anti-angiogenic activity.

## 3. Discussion

Angiogenesis plays a crucial role in the pathogenesis of angiogenesis-associated diseases, and control of angiogenesis may become a potential approach for the treatment of these diseases [22,23,24]. This study showed that E7050 effectively disrupted the key processes of angiogenesis, including the proliferation, migration, and tube formation of HUVECs. In addition, E7050 exhibited a striking inhibitory effect on in vivo angiogenesis of the CAM in chick embryos without any rupture of preexisting blood vessels. E7050 also suppressed tumor cell-induced angiogenesis and decreased tumor growth in a xenograft mouse model.

A large amount of VEGF is secreted from cancer cells at the early stage of tumor-induced angiogenesis and then binds to the transmembrane receptor VEGFR2 on endothelial cells [7]. Here, we detected VEGF expression in xenografted MES-SA/Dx5 tumor tissues via immunohistochemistry; however, there was no apparent inhibitory effect in E7050 treatment groups compared with the vehicle control group. At the same time, there was no obvious change in VEGF production in MES-SA/Dx5 cells with vehicle or E7050 treatment. These observations were likely because E7050 specifically influenced the function of endothelial cells but not tumor cells. VEGFR2 is the primary receptor in the VEGF signaling pathway, and phosphorylation of VEGFR2 at tyrosine residue 1175 is believed to be a key determinant of angiogenic signaling upon VEGF stimulation [21,25]. Furthermore, E7050 has been reported to diminish the phosphorylation of VEGFR2 at tyrosine residue 996 in xenografted KP-1/VEGF tumors [17]. Our results showed that treatment with E7050 significantly downregulated VEGF-induced expression levels of p-VEGFR2 (Tyr996 and Tyr1175) in HUVECs and notably suppressed p-VEGFR2 (Tyr1175) expression in the sections of xenografted MES-SA/Dx5 tumors, indicating E7050 acts as a potent VEGFR2 inhibitor.

Phosphorylation of VEGFR2 at Tyr1175 is required for the activation of PI3K/Akt, which regulates cell survival, proliferation, migration, protein synthesis, and angiogenesis [26,27]. It has been documented that downregulation of p-Akt in uterine sarcomas by anti-tumor inhibitors results in the reduction of cell proliferation and the induction of apoptosis [28,29]. PLCγ1, which interacts with VEGFR2 through phosphorylated Tyr1175, is an inducer of endothelial specification and has been reported to play an important role in cell proliferation and survival [30,31]. Activation of PLCγ1 induced by VEGF is also involved in regulating endothelial cell proliferation and tumor angiogenesis and growth [32]. In this study, we revealed that E7050 drastically suppressed VEGF-promoted phosphorylation of Akt and PLCγ1 in endothelial cells. E7050 also potently suppressed MES-SA/Dx5 CM-induced Akt activation in endothelial cells but had no inhibitory effect on the activation of PLCγ1.

VEGF has been reported to induce activation of Src kinase, which regulates cell migration and vascular permeability [33]. Phosphorylation of VEGFR2 induced by VEGF also leads to the activation of FAK, which participates in focal adhesion and regulates cell motility [34]. In the VEGF-mediated signaling pathway, the activated Src can form a complex with FAK, which initiates downstream signaling pathways and regulates cell proliferation, migration, and survival [35]. Previous studies have also indicated that inhibition of downstream protein kinases Src and FAK of VEGFR2 could be beneficial for cancer therapy [36,37]. In the current study, treatment with E7050 effectively suppressed the phosphorylation of Src and FAK in VEGF-stimulated endothelial cells. E7050 was also found to exert inhibitory effects on Src and FAK phosphorylation in MES-SA/Dx5 CM-stimulated endothelial cells.

In addition, growing evidence has shown that members of the MAPK family, including ERK1/2, JNK, and p38 MAPK, are involved in the regulation of cell proliferation, migration, and apoptosis [38]. The ERK1/2 pathway activated by VEGF has been implicated in the regulation of endothelial cell proliferation [39,40]. VEGF-induced p38 MAPK activation in HUVECs can induce rearrangements in the actin cytoskeleton that regulates cell migration [39]. Activation of p38 MAPK and JNK pathways is necessary for VEGF-induced angiogenesis [41]. A previous study also reported that VEGF-induced VEGFR2 phosphorylation in HUVECs initiates the activation of JNK; meanwhile, the activated JNK further stimulates VEGF-induced VEGFR2-sustained phosphorylation, which plays an essential role in VEGF-induced angiogenesis in HUVECs [42]. Moreover, Jun N-terminal kinase inhibitor (JNK-I) was found to inhibit the proliferation of human uterine carcinoma FU-MMT-1 cells and block FU-MMT-1-stimulated human arterial endothelial cell (HAEC) tube formation, suggesting that JNK-I is an anti-angiogenic agent [43]. In the present study, we found that E7050 significantly suppressed the phosphorylation of p38 MAPK and JNK in VEGF- or MES-SA/Dx5 CM-stimulated endothelial cells. However, E7050 prompted the phosphorylation of ERK1/2 in VEGF- or MES-SA/Dx5 CM-stimulated endothelial cells. By integrating the data obtained, a schematic model of the mechanisms of the E7050-induced anti-angiogenic effect in endothelial cells was developed, as shown in Figure 7. The inhibitory actions of E7050 on endothelial cell proliferation and migration, as well as angiogenesis, are proposed to be associated with the suppression of VEGF-induced autophosphorylation of VEGFR2 and the decreased activation of VEGFR2-mediated signaling pathways. Moreover, our previous study has shown that E7050 exerted anti-cancer activity in both MES-SA/Dx5 cells and an MES-SA/Dx5 xenograft mouse model [44].

## 4. Materials and Methods

### 4.1. Chemicals and Reagents

Medium 199 (M199), recombinant human VEGF, fetal bovine serum (FBS), sodium pyruvate, and penicillin–streptomycin were purchased from Thermo Fischer Scientific (Waltham, MA, USA). Bovine serum albumin (BSA), McCoy’s 5A medium, dimethyl sulfoxide (DMSO), and doxorubicin (DOX) hydrochloride were purchased from Sigma-Aldrich (Saint Louis, MO, USA). E7050 (Golvatinib) (purity: 99.78%) was obtained from Selleckchem (Houston, TX, USA). E7050 was dissolved in DMSO and stored at −20 °C in a refrigerator. Primary antibodies against VEGFR2 (#2479; 1:20,000 dilution), phospho (p)-VEGFR2 (Tyr1175) (#2478; 1:1500 dilution), Akt (#4691; 1:20,000 dilution), p-Akt (Ser473) (#4060; 1:5000 dilution), FAK (#3285; 1:7000 dilution), p-FAK (Tyr397) (#8586; 1:1500 dilution), PLCγ1 (#5690; 1:8000 dilution), p-PLCγ1 (Ser1248) (#8713; 1:2500 dilution), Src (#2109; 1:30,000 dilution), p-Src (Tyr416) (#6943; 1:6000 dilution), JNK (#9258; 1:1000 dilution), p-JNK (Thr183/Tyr185) (#4668; 1:1000 dilution), ERK1/2 (#4695; 1:15,000 dilution), p-ERK1/2 (Thr202/Tyr204) (#4370; 1:15,000 dilution), p38 MAPK (#9212; 1:12,000 dilution), and p-p38 MAPK (Thr180/Tyr182) (#4511; 1:2000 dilution) were purchased from Cell Signaling Technology (Beverly, MA, USA). Antibodies against VEGF (sc-7269; 1:300 dilution), HSP27 (sc-13132; 1:2000 dilution), and p-HSP27 (Ser82) (sc-166693; 1:30,000 dilution) antibodies were purchased from Santa Cruz Biotechnology (Dallas, TX, USA). An antibody specific for CD31/PECAM1 (A2104; 1:200 dilution) was obtained from ABclonal Technology (Woburn, MA, USA). An antibody specific for p-VEGFR2 (Tyr996) (ab135776; 1:1500 dilution) was obtained from Abcam (Cambridge, MA, USA). The β-actin (20536-1-AP; 1:80,000 dilution) antibody was obtained from Proteintech (Chicago, IL, USA). Horseradish peroxidase (HRP)-conjugated goat anti-rabbit (sc-2004; 1:20,000 dilution) and anti-mouse (sc-2005; 1:20,000 dilution) secondary antibodies were obtained from Santa Cruz Biotechnology.

### 4.2. Cell Culture

Human umbilical vein endothelial cells (HUVECs, BCRC H-UV001) were obtained from the Bioresource Collection and Research Center (BCRC, Hsinchu, Taiwan). Human uterine sarcoma cell line MES-SA (CRL-1976) was purchased from the American Type Culture Collection (ATCC, Manassas, VA, USA). The multidrug-resistant human uterine sarcoma cell line MES-SA/Dx5 was established from MES-SA cells in the presence of increasing DOX concentrations, as described previously [45]. HUVECs were cultured in M199 medium containing the endothelial growth medium (EGM)-2 SingleQuots Kit (consisting of FBS, VEGF and other growth factors) (Lonza, Basel, Switzerland), 1 mM sodium pyruvate, 10 mM HEPES, 100 U/mL penicillin, and 100 μg/mL streptomycin. MES-SA/Dx5 cells were cultured in McCoy’s 5A medium supplemented with 10% FBS, 1 mM sodium pyruvate, 100 U/mL penicillin, 100 μg/mL streptomycin, and 0.4 μg/mL of DOX. All cell lines were maintained at 37 °C in a humidified incubator at 5% CO_2_.

### 4.3. Cell Viability Assay

The effect of E7050 on the viability of HUVECs was tested using the 3-(4,5-dimethylthiazol-2-yl)-2,5-diphenyltetrazolium bromide (MTT) assay, as previously mentioned [46]. Briefly, a density of 2 × 10^4^ HUVECs/well was seeded into 96-well plates and cultivated under normal cell culture conditions for 24 h. The cells were then treated with the indicated concentrations of E7050 (5, 10, and 25 μM) or vehicle (1% DMSO) and incubated for 6 or 24 h. All reagents were added according to the manufacturer’s instructions.

### 4.4. Wound Healing Migration Assay

A density of 5 × 10^5^ HUVECs/well was seeded in 6-well plates. After 24 h of incubation, the cells were starved with M199 basal medium for 6 h. The monolayer cells were wounded by being scratched along a straight line with a sterile 200 μL pipette tip and then gently washed three times with phosphate buffered saline (PBS) to remove the detached cells. M199 medium supplemented with EGM-2 SingleQuots and different concentrations of E7050 (2.5, 5, and 10 μM) was added to the wells. After 16 h incubation at 37 °C, the viable cells were detected using calcein AM (Invitrogen, Carlsbad, CA, USA). Images of cells were measured at 485/538 nm (ex/em) using a fluorescence microscope. The migration of HUVECs into the scrape wound was observed in each group and quantified by counting.

### 4.5. Endothelial Cell Tube Formation Assay

Matrigel basement membrane matrix (BD Biosciences, San Jose, CA, USA) was thawed overnight at 4 °C. Each well of Ibidi μ-slides (Ibidi GmbH, Munich, Germany) was coated at 4 °C with 10 μL Matrigel and subsequently allowed to polymerize at 37 °C for 30 min. A suspension of HUVECs (1 × 10^4^ cells) was seeded onto each Matrigel-coated well and then treated with vehicle (1% DMSO) or E7050 (5, 10, and 25 μM). After incubation at 37 °C for 6 h, the viable cells were detected using calcein AM (Invitrogen). Images of the capillary-like tubular structures of endothelial cells in each well were photographed using a fluorescence microscope, and the extent of tube formation was evaluated by measuring total tubule area per field.

### 4.6. Chick Embryo Chorioallantoic Membrane (CAM) Assay

For the in vivo anti-angiogenesis activity of E7050, the chick embryo CAM assay was performed as previously described [47]. Briefly, fertilized chicken eggs were maintained in a humidified egg incubator (80% humidified atmosphere) at 37 °C. After 3 days of incubation, a small hole was made using an 18-gauge hypodermic needle to extract 6–8 mL of albumin from the egg. An approximately 1 × 1 cm^2^ window in the upper eggshell was carefully created to check the healthy and normal growth of embryos and to exclude malformations or dead embryos. After fenestration, the windows were sealed with tape to prevent contamination and eggs were placed back into the incubator. On day 9 of incubation, different dosages of E7050 (5, 10, and 25 μM) diluted in 20 μL of 50% Matrigel (BD Biosciences) were directly added on the CAM through the small window created before. All the above procedures were performed under aseptic conditions. After 48 h of treatment, each egg was observed, and blood vessels were photographed. Angiogenesis was quantified by counting total vessel network length, which was estimated using WimCAM software (Onimagin Technologies, Córdoba, Spain).

### 4.7. Enzyme-Linked Immunosorbent Assay (ELISA)

The concentrations of VEGF in the culture supernatants of E7050-treated MES-SA/Dx5 cells were measured using the commercial human VEGF ELISA Kit PicoKine^TM^ according to the instructions of the manufacturer (Boster Biological Technology, Pleasanton, CA, USA).

### 4.8. Tumor Xenograft Model and Immunohistochemical Analysis

Human MES-SA/Dx5-LG cells (1 × 10^6^ cells) were mixed with Matrigel (BD Biosciences) and then subcutaneously injected into the dorsal region near the thigh of the male BALB/c nude mice. When the xenograft tumor size reached a volume of approximately 50 mm^3^, the mice were divided randomly into three groups (*n* = 5 per group) and administered orally with vehicle (distilled water) or E7050 (50 mg/kg and 175 mg/kg) once daily for 28 days. All mice were sacrificed at the end of treatment; the tumors were then dissected and collected, fixed in 4% paraformaldehyde for 72 h, embedded in paraffin, and cut into sections. The paraffin sections (5 μm) were deparaffinized in xylene and dehydrated in graded alcohol before finally being hydrated in water. Antigen retrieval was performed by boiling the slides in 10 mM sodium citrate buffer (pH 6.0) for 10 min. Endogenous peroxidase activity was quenched with 3% hydrogen peroxide. Immunohistochemical staining of deparaffinized sections was performed using the EnVision Detection System (Dako, Glostrup, Denmark) according to the manufacturer’s instructions. Tumor tissue sections were incubated with specific primary antibodies against CD31, VEGF, or p-VEGFR2 (Tyr1175) overnight at 4 °C. Finally, the sections were stained with 3,3′-diaminobenzidine (DAB), counterstained with hematoxylin, dehydrated, and mounted. Two expert pathologists observed and evaluated the staining results of all slices in a blind manner. Images from stained slices were taken using HistoFAXS (Tissue Gnostics, Vienna, Austria). The animal experimental protocols were reviewed and approved by the Institutional Animal Care and Use Committee of Chang Gung University (Taoyuan, Taiwan).

### 4.9. Western Blot Analysis

Protein extraction and Western blot analysis were performed as described previously [42]. In brief, equal amounts of protein samples were subjected to 8–12% sodium dodecyl sulfate-polyacrylamide gel electrophoresis (SDS-PAGE) and transferred electrophoretically onto polyvinylidene difluoride (PVDF) membranes (0.45 μm; Millipore, Billerica, MA, USA). The membranes were blocked in TBST buffer (0.1% Tween-20 in 1 × Tris buffered saline, pH 7.4) containing 5% *w*/*v* skim milk or BSA for 1 h at room temperature, followed by incubation with the appropriate primary antibodies overnight at 4 °C. After washing steps, the membranes were incubated with the corresponding HRP-conjugated secondary antibodies for 1.5 h at room temperature. Finally, immunoreactive bands were detected using an enhanced chemiluminescence (ECL) reagent (Millipore) and quantified by densitometry. β-actin was used as a loading control.

### 4.10. Statistical Analysis

All data values are presented as mean ± standard error of the mean (SEM) from at least three independent experiments. Differences between two groups were evaluated using the two-tailed Student’s *t*-test. Comparisons for multiple groups were applied using one-way analysis of variance (ANOVA) followed by Dunnett’s post hoc test. The results were considered statistically significant when the *p* value was less than 0.05.

## 5. Conclusions

In summary, our study demonstrates the anti-angiogenic action and corresponding mechanisms of E7050 in vitro, as well as the therapeutic effect of multidrug-resistant uterine sarcoma in vivo. E7050 treatment not only significantly decreased proliferation, migration, and capillary-like tube formation in endothelial cells, but also potently repressed neovascularization in the chick embryo CAM model. The mechanistic effect of E7050-induced angiogenesis inhibition was due to pronounced suppression of the activation of VEGFR2 and its downstream signaling effectors. Furthermore, the expression levels of CD31 and p-VEGFR2 were inhibited in the tumor tissues of E7050-treated xenografted MES-SA/Dx5 tumor mice, suggesting that E7050 suppressed xenografted MES-SA/Dx5 tumor growth at least partially owing to anti-angiogenesis. E7050 also inhibited the activation of VEGFR2 and its downstream signaling effectors in MES-SA/Dx5 CM-stimulated endothelial cells. Therefore, E7050 may act as a multifunctional anti-cancer agent through its inhibitory effect on pivotal aspects of both tumor growth and angiogenesis.

## Figures and Tables

**Figure 1 ijms-24-09606-f001:**
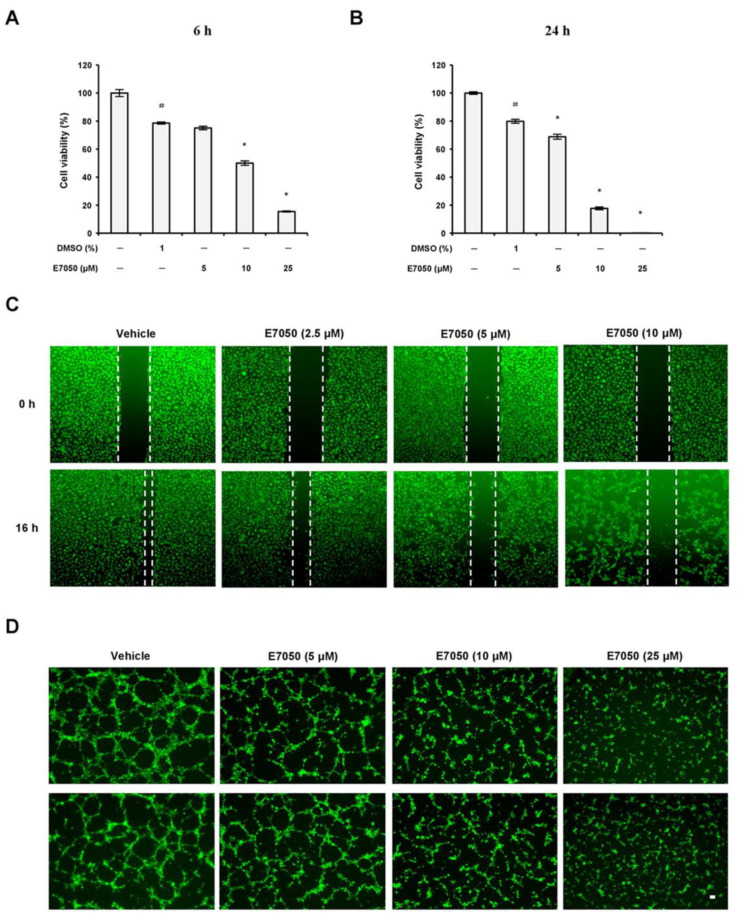
Effects of E7050 on proliferation, migration, and capillary-like tube formation in human endothelial cells. HUVECs were treated with vehicle control or various concentrations of E7050 (5, 10, and 25 μM) for (**A**) 6 h or (**B**) 24 h, after which the extent of cell proliferation was determined by an MTT assay. The results are expressed as the percentage of cell viability relative to the untreated control. (**C**) The wound healing migration assay. The cell monolayer was scratched by a pipette and then treated with vehicle or E7050 at indicated concentrations. After 16 h incubation, migration of cells was observed and then images were taken through a microscope. (**D**) The tube formation assay. The cells were seeded in Matrigel-coated Ibidi μ-slides and then treated with different concentrations of E7050. After 6 h incubation, the capillary-like structures were observed and photographed. Scale bar: 50 μm. (**E**) Cell migration levels in HUVECs following E7050 treatment were quantified and expressed as a percentage relative to the vehicle control. (**F**) Tube formation after treatment with E7050 was measured and expressed as a percentage by normalization with the vehicle control. All data are presented as mean ± SEM of three independent experiments. # *p* < 0.05 compared with the untreated cells. * *p* < 0.05 compared with the vehicle-treated cells.

**Figure 2 ijms-24-09606-f002:**
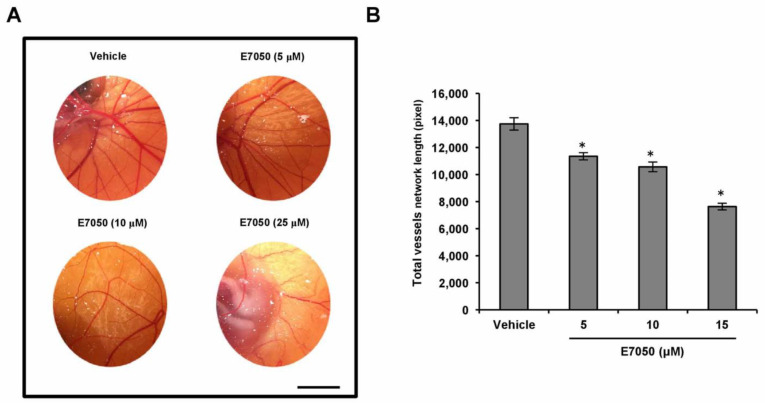
Effect of E7050 on in vivo angiogenesis using the chick embryo CAM assay. (**A**) Representative images of new blood vessel formation on the in vivo CAM model after exposure to various concentrations of E7050 (5–25 μM) are shown. Scale bar: 1 cm. (**B**) The bar chart represents the network length of total blood vessels in different treatment groups. The data are presented as mean ± SEM (*n* = 3 in each group). * *p* < 0.05 compared with the vehicle-treated cells.

**Figure 3 ijms-24-09606-f003:**
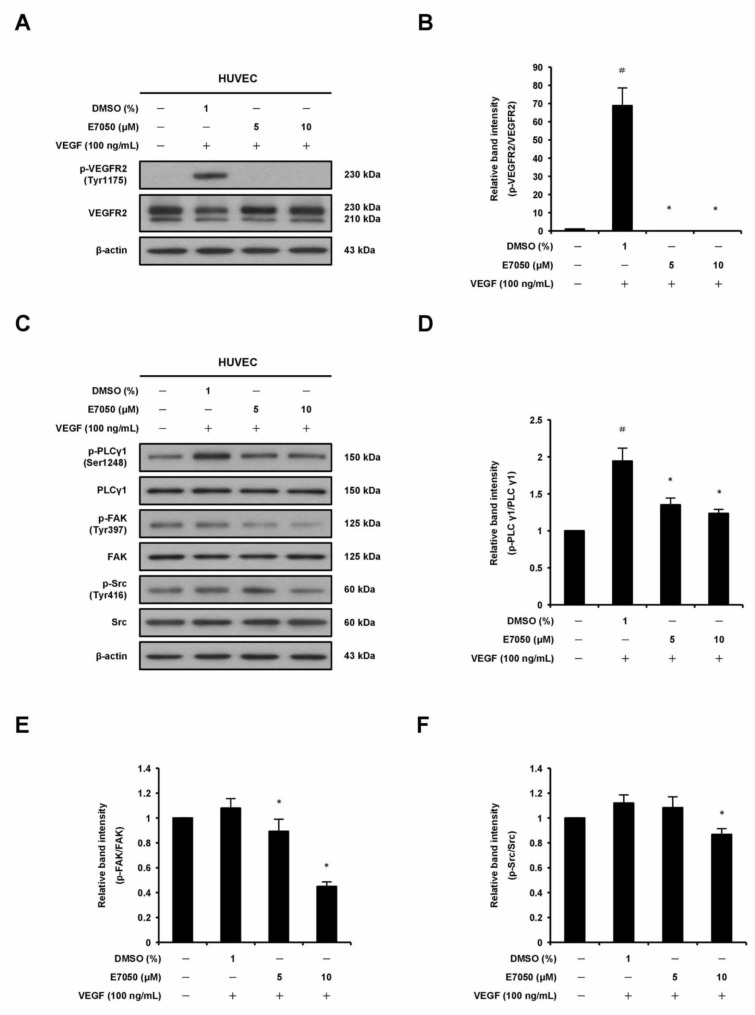
Effects of E7050 on VEGFR2-mediated signaling pathways in VEGF-stimulated HUVECs. Cells were serum-starved for 6 h and then pretreated with E7050 (5 and 10 μM) or vehicle for 1 h, followed by stimulation with VEGF (100 ng/mL) for another 10 min (VEGFR2, PLCγ1, FAK, and Src) or 30 min (Akt, JNK, and p38 MAPK) before proteins were collected. The expression and phosphorylation status of VEGFR2 and its downstream effectors, including PLCγ1, FAK, Src, Akt, JNK, and p38 MAPK, were detected by Western blotting using respective antibodies. (**A**) E7050 inhibited the phosphorylation (Tyr1175) of VEGFR2 induced by VEGF in HUVECs. (**B**) The quantified results show that the ratio of p-VEGFR2 protein normalized to the total amount of VEGFR2 protein, which was measured by densitometry. (**C**) E7050 inhibited the phosphorylation of PLCγ1, FAK, and Src in VEGF-stimulated HUVECs. Calculated ratios of (**D**) p-PLCγ1, (**E**) p-FAK, and (**F**) p-Src normalized to the relative total protein levels are shown. (**G**) E7050 inhibited the phosphorylation of Akt, JNK, and p38 MAPK in VEGF-stimulated HUVECs. The compiled results of the ratios of (**H**) p-Akt, (**I**) p-JNK, and (**J**) p-p38 MAPK normalized to the relative total protein levels are shown. The data are presented as mean ± SEM of three independent experiments. # *p* < 0.05 compared with the untreated cells. * *p* < 0.05 compared with the vehicle-treated cells.

**Figure 4 ijms-24-09606-f004:**
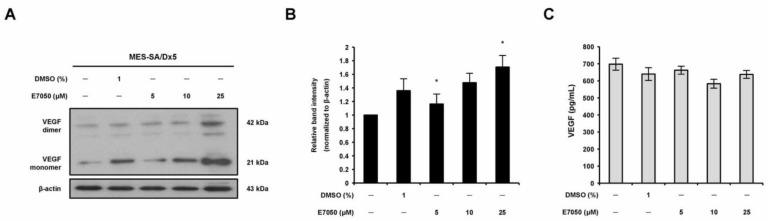
Effect of E7050 on the expression level of HGF in MES-SA/Dx5 cells. (**A**) Cells were treated with various concentrations (5–25 μM) of E7050 for 24 h. Whole cell extracts were prepared and subjected to Western blotting using antibodies against HGF and β-actin. β-actin was used as an internal loading control. (**B**) Densitometric analysis of blots relative to HGF protein after normalization with β-actin. (**C**) Secreted HGF in cell culture media was determined by ELISA. Data are presented as the mean ± SEM of three independent experiments. * *p* < 0.05 versus vehicle-treated control cells.

**Figure 5 ijms-24-09606-f005:**
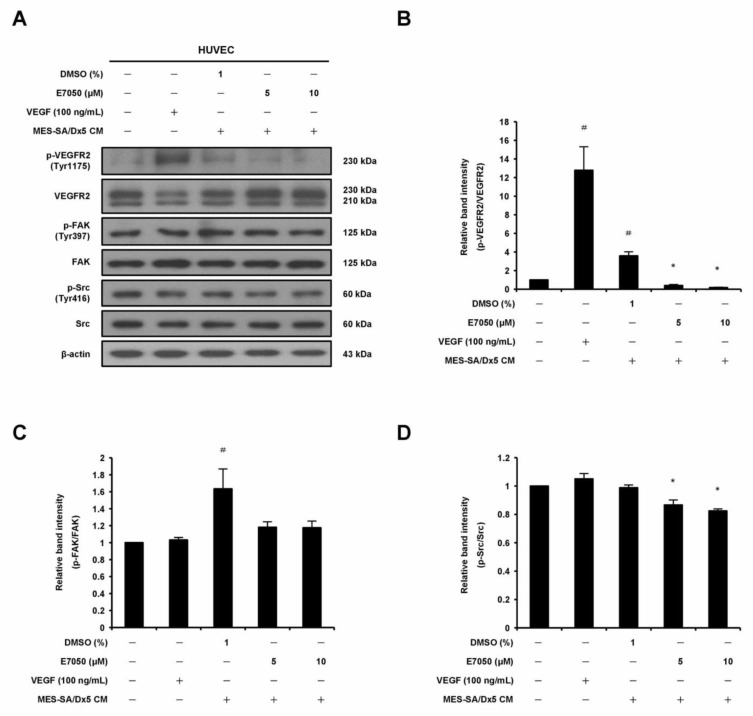
Effects of E7050 on VEGFR2-mediated signaling pathways in cultured MES-SA/Dx5 cells-derived conditioned medium (CM)-treated HUVECs. Cells were serum-starved for 6 h and pretreated with E7050 (5 and 10 μM) or vehicle for 1 h, followed by the addition of VEGF (100 ng/mL) or CM (from cultured MES-SA/Dx5 cells) for another 10 min (VEGFR2, FAK, and Src) or 30 min (Akt, JNK, and p38 MAPK) before protein extraction. The expression and phosphorylation status of VEGFR2 and its downstream effectors, including FAK, Src, Akt, JNK, and p38 MAPK, were detected by Western blotting using specific antibodies. (**A**) E7050 inhibited the phosphorylation of VEGFR2 and Src in MES-SA/Dx5 CM-induced HUVECs. (**B**) The relative band density of p-VEGFR2 protein was normalized to total VEGFR2 protein, which was measured by densitometry. Calculated ratios of (**C**) p-FAK and (**D**) p-Src normalized to the relative total protein levels are shown. (**E**) E7050 inhibited the phosphorylation of Akt, JNK, and p38 MAPK in MES-SA/Dx5 CM-induced HUVECs. The compiled results of the ratios of (**F**) p-Akt, (**G**) p-JNK, and (**H**) p-p38 MAPK normalized to relative total protein levels are shown. The data are presented as mean ± SEM of three independent experiments. # *p* < 0.05 compared with the untreated cells. * *p* < 0.05 compared with the vehicle-treated cells.

**Figure 6 ijms-24-09606-f006:**
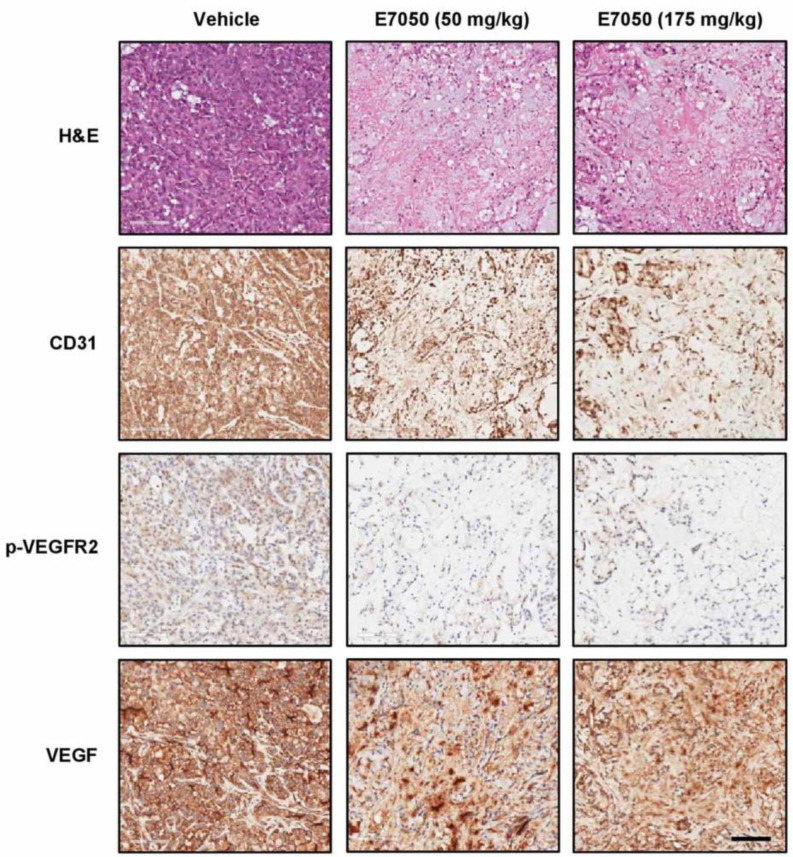
Effects of E7050 on tumor growth and angiogenesis in the MES-SA/Dx5 cell line-derived xenograft mouse model. Histological characteristics in the tumor tissue sections of MES-SA/Dx5 xenografts obtained from vehicle- and E7050-treated nude mice on day 28 were measured by H&E staining. The expression levels of CD31, VEGF, and p-VEGFR2 (Tyr1175) in tumor tissue sections were also examined by immunohistochemical analyses. Representative photomicrographs of H&E and immunohistochemical staining in tumor tissue sections from vehicle control and E7050-treated groups of mice are shown. Scale bar: 100 μm.

**Figure 7 ijms-24-09606-f007:**
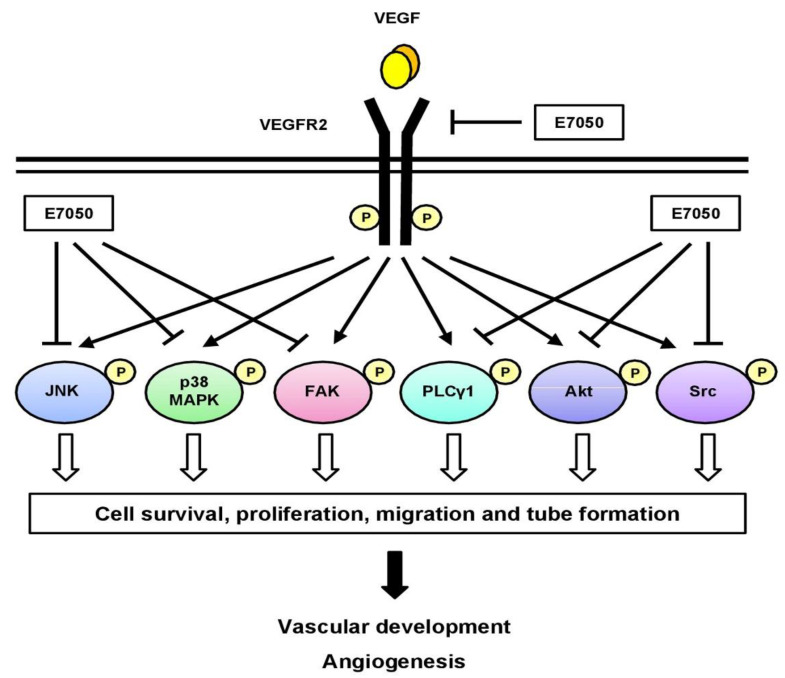
Schematic diagram of a proposed mechanism of E7050-induced anti-angiogenic activity. E7050 exerts anti-angiogenic effects in VEGF-stimulated endothelial cells by downregulating the phosphorylation of VEGFR2 and its downstream mediators, including PLCγ1, FAK, Src, Akt, JNK, and p38 MAPK. The blockage of VEGFR2-mediated signaling cascade pathways by E7050 contributes to the inhibition of proliferation, migration, and tube formation in endothelial cells.

## Data Availability

The data presented in this study are available on request from the corresponding author.

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
