# Peer review of "E7050 Suppresses the Growth of Multidrug-Resistant Human Uterine Sarcoma by Inhibiting Angiogenesis via Targeting of VEGFR2-Mediated Signaling Pathways"

_ijms, 2023, doi:10.3390/ijms24119606_

Round 1

Reviewer 1 Report

The authors evaluated the anti-angio-28 genic activity of E7050 in vitro and in vivo, and define the underlying molecular mechanism. The study is well designed and represented.

Reviewer 2 Report

 he manuscript entitled “E7050 Suppress the growth of Multidrug-Resistant Human Uterine Sarcoma by Inhibiting Angiogenesis Via Targeting the VEGFR2-Mediated Signalling pathwaysby Tsung Teng Huang et. al. describes the antiangiogenic activity of E7050 in vitro and In vivo.  The manuscript can be accepted once the below comments are addressed.

Comments:

1.     Figure 1D, what does the two panel represents? Are they represents two time points of experiment?

2.     In Figure 1C, 1D and 2A, the text is clipped figure please correct.

3.     Authors should compare the E7050 with already reported inhibitors (as positive control) to support the molecular mechanism of E7050 for all experiements.

4.     What the advantages and disadvantages of known inhibitors vs E7050 should be discussed (anti angiogenic activity via VEGFR2)

5.     The time point are different for Cell viability, migration and Wound healing assay.  What happen to HUVECs cells at 16 hr drug treatment.It is suggested to include MTT assay at 16 hr.

6.     In western blot Analysis, please mention the PVDF pore size

7.     Combinatorial treatment

8.     A p-VEGFR2 blot should be included for Figure 3C, 3G, and 5E

9.     The p38 phosphorylation is inhibited by E7050 at 10 uM in Figure 3G, But in Figure 5E E7050 fails inhibit (at same concentration) completely why?

10.  To support the inhibitory activity of E7050 against Tyr 1175. A point mutation of Tyr 1175 in the presence of inhibitor will be helpful to support the mechanistic studies.

11.  Can we use Src and FAK kinase inhibitors along with E7050 part of combinatorial therapy?

12.   Why HUVECs cell lines develop multidrug resistance (MDR), how E7050 overcome the MDR is not discussed.

13.  Please mention seeding density for wound healing and migration assays

14.  Please provide catalog details for antibodies and dilution used for IHC, western blot etc

15.   Line 439, include percentage of paraformaldehyde used and incubation time.

16.  Screening of E7050 against panel of 518 human kinases will be helpful to address the specificity and off target activity of E7050 

Reviewer 3 Report

Title: E7050 Suppresses the Growth of Multidrug-Resistant Human 2 Uterine Sarcoma by Inhibiting Angiogenesis via Targeting the 3 VEGFR2-Mediated Signaling Pathways

Corresponding author: Kowit-Yu Chong

This is a very interesting manuscript. However, to be published, it is recommended to study the following issues.

1-    The authors should add the sources and purity percentage for all chemical compounds used in the current study.

2-    Improve the quality and resolution for all figures.

3-    Transfer Fig. 6 from the discussion section to the results section.

4-    In page 2 line 68-70, the authors should state the strategy of cancer prevention and treatment through inhibition of the VEGF and angiogenesis by using of chemotherapeutic drugs via adding the following reference: https://doi.org/10.1016/j.intimp.2014.05.007

5-    Transfer reference no. 44 with its findings from the conclusion section to the discussion section.

6-    Graphical abstract is highly recommended.

7-    Please adjust all the references according to the journal’s instructions.

Overall, the manuscript can be considered for publication after minor revision as indicated above.

Author Response

Please see the attchment
